# Development of Sound Localization in Infants and Young Children with Cochlear Implants

**DOI:** 10.3390/jcm11226758

**Published:** 2022-11-15

**Authors:** Filip Asp, Eva Karltorp, Erik Berninger

**Affiliations:** 1Department of Clinical Science, Intervention and Technology, Karolinska Institutet, 17177 Stockholm, Sweden; 2Hearing Implant Unit, Department of ENT, Karolinska University Hospital, 14186 Stockholm, Sweden; 3Department of Audiology and Neurotology, Karolinska University Hospital, 14186 Stockholm, Sweden

**Keywords:** sound localization, infants, bilateral cochlear implants, development

## Abstract

Cochlear implantation as a treatment for severe-to-profound hearing loss allows children to develop hearing, speech, and language in many cases. However, cochlear implants are generally provided beyond the infant period and outcomes are assessed after years of implant use, making comparison with normal development difficult. The aim was to study whether the rate of improvement of horizontal localization accuracy in children with bilateral implants is similar to children with normal hearing. A convenience sample of 20 children with a median age at simultaneous bilateral implantation = 0.58 years (0.42–2.3 years) participated in this cohort study. Longitudinal follow-up of sound localization accuracy for an average of ≈1 year generated 42 observations at a mean age = 1.5 years (0.58–3.6 years). The rate of development was compared to historical control groups including children with normal hearing and with relatively late bilateral implantation (≈4 years of age). There was a significant main effect of time with bilateral implants on localization accuracy (slope = 0.21/year, R^2^ = 0.25, F = 13.6, *p* < 0.001, *n* = 42). No differences between slopes (F = 0.30, *p* = 0.58) or correlation coefficients (Cohen’s q = 0.28, *p* = 0.45) existed when comparing children with implants and normal hearing (slope = 0.16/year since birth, *p* = 0.015, *n* = 12). The rate of development was identical to children implanted late. Results suggest that early bilateral implantation in children with severe-to-profound hearing loss allows development of sound localization at a similar age to children with normal hearing. Similar rates in children with early and late implantation and normal hearing suggest an intrinsic mechanism for the development of horizontal sound localization abilities.

## 1. Introduction

Humans can identify the source of a sound with high accuracy [1]. Interaural differences in time and level are analyzed in the central auditory system and associated with events or locations in our environment. Even though both animals [2,3] and humans [4] can localize sound just after birth, accuracy refines with experience from such associations [5,6]. In barn owls, an extensively studied species, the visual system plays a key role for the brain to learn and build an auditory space map based on these associations [7,8]. Occlusion of one ear [9] or displacement of the visual field [10] in the barn owl have shown corrections of sound localization behavior in response to these manipulations. These corrections in localization behavior are experience-driven and demonstrated using various experimental manipulations of sensory input in many species. Experience gained early in life is demonstrated to be most important for the formation and refinement of a subcortical auditory space map [11,12,13], but capability of adaptation to altered cues is shown behaviorally in adult humans [6,14,15] and ferrets [16]. Plasticity in the neural circuitry underlying sound localization, thus, exists across species and age (see [16] for an overview).

Children with congenital bilateral severe-to-profound hearing loss represent an opportunity for the study of plasticity in the human auditory system. For these children, cochlear implantation is a clinically well-established treatment resulting in an ability to recognize speech and development of speech and language in many cases [17,18,19,20]. To promote normal speech and language development, implantation preferably should occur no later than 12 months of age [21,22] and family centered early intervention is important. However, horizontal sound localization, an important and early obtainable auditory ability dependent on bilateral implantation [17], develops systematically despite relatively late sequential bilateral cochlear implantation (≈4 years of age) following long periods of unilateral hearing (≈2 years) [23]. Once bilateral stimulation is provided, development of sound localization accuracy occurs over several years [23], with subsequent persistent abilities [24], albeit, worse than normal [17,24]. Accordingly, the age at which implants are provided does not seem to determine development of sound localization [23], consistent with findings that adult humans can adapt to altered localization cues [6,25].

While a number of large centers perform cochlear implantation early, implants are generally provided beyond the infant period [26,27] and the US Food and Drug administration grants implantation no earlier than 0.75 years (for one of three major manufacturers). Relatively late implantation in combination with assessment of behavioral outcomes after years of implant use makes comparison with normal developmental trajectories difficult.

Here, we study development of horizontal sound localization in infants and young children listening through bilateral cochlear implants (BiCI) since ≈0.6 years of age, and contrast the results with previous findings from children with normal hearing [28], and children with late cochlear implantation [23]. We asked whether early bilateral cochlear implantation allows experience-driven improvement of horizontal localization accuracy and if the rate of improvement was similar to children with normal hearing.

## 2. Materials and Methods

### 2.1. Study Design

This was a longitudinal clinical study with an inclusion period between March 2019 and February 2021. The study was approved by the regional ethical review board in Stockholm, Sweden (permit number 2012/189-31/3 and 2013/2248-32). To be included, children were required to have received bilateral cochlear implantation in a simultaneous procedure and be available and willing to participate in the study during regular clinical follow-up. Within 3 months after surgery, parents were asked for their child’s participation at a visit to the clinic. At clinical follow-ups (initial fitting of external parts of the cochlear implant system about 3 weeks after surgery, and then approximately 1, 3, 6, 12, 18 and 24 months after initial fitting), children participated in a 3 min horizontal sound localization task adapted to children from about 6 months of age [28]. The rate of development of sound localization accuracy was compared to children with normal hearing and older children using cochlear implants.

### 2.2. Subjects

Twenty children were included in the study at a median age of 0.87 years (0.58–2.53 years, 8 females) (Table 1). Parental informed consent was obtained for all children. Children were implanted bilaterally at a median age of 0.58 years (0.42–2.3 years) with devices from Cochlear (Cochlear Corporation, Sydney, Australia) or Med-El (Med-El GmbH, Innsbruck, Austria).

Thirteen children who met the inclusion criteria were not asked to participate due to limited time during regular clinical follow-up. Another two children declined participation. These 15 children were implanted at the same median age as the included subjects.

### 2.3. Setup, Stimulus and Test Procedure

The setup, stimulus, test procedure and acquisition of behavioral responses is described in detail previously [28]. Children were seated in the lap of a parent in front of 12 active loudspeakers (ARGON 7340A; Argon Audio, Sweden) spanning a 110-degree arc in the frontal horizontal plane (Figure 1) in an audiometric test room. Loudspeakers were at ear level and spaced 10 degrees, resulting in loudspeaker positions at ±55, ±45, ±35, ±25, ±15, and ±5 degrees azimuth with respect to the subject. A 7-inch thin film transistor (TFT) display was mounted below each loudspeaker, resulting in 12 loudspeaker/display (LD)-pairs. An eye-tracking system (Smart Eye Pro; Smart Eye AB, Gothenburg, Sweden) was used for objective positioning of children’s pupil positions relative to the LD-pairs.

A sound localization test consisted of 24 azimuthal shifts of an ongoing auditory-visual stimulus (a colorful cartoon movie playing a continuous melody with a long-term frequency spectrum similar to female speech) presented at 63 dBA. In each azimuthal shift, the sound was changed to another randomly assigned loudspeaker on average every 7th second (5–9 s) with a simultaneous stop of the visual stimulus. The visual stimulus was reintroduced on the visual display corresponding to the sounding loudspeaker 1.6 s after the azimuthal sound shift. The procedure allowed acquisition of gaze behavior during 1.6 s in response to a spatial change of the sound. A test lasted ≈3 min.

Localization accuracy was quantified by an Error Index [29,30]. An EI = 0 corresponds to perfect performance. An EI = 1 corresponds to average random performance. A Monte Carlo simulation showed that the 95% confidence interval (C.I.) for random performance using the current procedure was [0.72, 1.28].

Children were not given any instructions before or during testing. The parent having the child on their knee was instructed to remain seated and unmoving and to not talk to the child.

### 2.4. Analyses

To analyze cross-sectional data, linear regression analyses of the first sound localization test (*n* = 18; two children were not possible to assess) were performed with EI (range = 0.31–1.0) as dependent variable and time since activation of BiCI (i.e., auditory experience, range = 0.03–1.7 years) as the independent variable. To account for between- and within-subject variability despite missing data points, a linear mixed model was constructed, with the EI as dependent variable and time since activation of BiCI, age at implantation, and the number of obtained responses in a localization test as fixed effects. A random intercept for subjects was included in the model and interaction terms between fixed effects and random intercepts were evaluated. Selection of a final model was guided by minimizing the Aikaike information criterion. The slope of the regression line was statistically compared to slopes obtained in children with normal hearing [28] by an analysis of covariance, and qualitatively to older children with cochlear implants [23]. Statistical analyses were performed using Statistica version 13 (Statsoft, Inc., Tulsa, OK, USA) and R Version 3.4.2 (R Foundation for Statistical Computing, Austria)

Test reliability was computed by dividing each test into two parts and comparing the EI between part 1 (test) and part 2 (retest) [28]. The statistical reliability of the localization test was then quantified by analysis of the variability in test–retest differences and by estimation of the variance in EI for a single SLA measurement (see Equation (10) in [28] for the variance estimate).

## 3. Results

Longitudinal follow-up generated 42 sound localization measurements (1 to 5 measurements per child) in 18 children (2 of 20 children were not willing to cooperate to testing). The average time since activation of BiCI was 0.9 years (0.09 years–1.7 years, *n* = 42) with a mean age at test = 1.5 years (0.58 years–3.6 years, *n* = 42).

### 3.1. Development of Sound Localization Accuracy Is Driven by Time since Activation of Bilateral Implants

Simple linear regression analyses of cross-sectional data (first test, *n* = 18) indicated that increasing time since activation of BiCI was associated with increasing sound localization accuracy (EI = 0.83–0.19 time since BiCI, r = −0.47, *p* = 0.05). A linear mixed model for the entire longitudinal dataset (*n* = 42) showed a significant main effect of time since BiCI on the EI, whereas no effect of random intercept (i.e., of subject) or interaction with number of trials existed. The final linear model, thus, included time since activation of BiCI, which explained 25% of the variance in the EI (R^2^ = 0.25, F = 13.6, *n* = 42, *p* = 0.0007) (Figure 2). According to the model equation, which was similar to the linear fit from the cross-sectional analysis, the EI decreased by 0.21/year with an intercept of 0.82.

Individual perceived versus presented azimuths were plotted for the child with the longest time since activation of BiCI (Figure 3), to visualize development of behavioral response patterns. With increasing time since activation of BiCI (4 measurements over 1.4 years follow-up), perceived azimuths were approaching target azimuths.

### 3.2. Comparison between Children with Early Bilateral Cochlear Implantation and Young Children with Normal Hearing

To study whether implanted children develop sound localization accuracy similar to children with normal hearing, data were compared with previously reported cross-sectional results from 12 children (median age = 1.0 years) with normal hearing tested with the same technique [28]. Figure 4, panel A, illustrates that children with implants (black open circles) overlap in their performance with children with normal hearing (filled blue circles).

The slopes of the regression lines for each group were similar (Normal hearing: 0.16/year, *p* = 0.015); Cochlear implant: 0.21/year, *p* = 0.0007) (Figure 4, panel B), with no difference between correlation coefficients (Cohen’s q = 0.28, *p* = 0.45). An analysis of covariance with group as categorical factors (cochlear implants versus normal hearing) and time since bilateral hearing onset/age as a covariate, showed no statistically significant interaction (F = 0.30, *p* = 0.58), that is, no significant difference between developmental rates. In addition, no significant difference in localization accuracy existed between children with implants and normal hearing (F = 3.0, *p* = 0.09).

### 3.3. Comparison between Children with Early Bilateral Cochlear Implantation and Relatively Late Sequential Bilateral Cochlear Implantation

To study the effect of age at implantation on development of sound localization accuracy, data were further compared to results from children with relatively late sequential bilateral cochlear implantation (median age first CI = 1.9 years, median age second CI = 4.1 years, *n* = 66) [23]. These children, implanted and tested at the same tertiary referral center as the subjects in the present study, were assessed at a median age of 5.6 years (range = 2.8–17.3 years), i.e., they were substantially older than the children in the present study. Despite methodological and procedural differences (i.e., number of sound-sources, spatial range and resolution of the test, spectral and temporal characteristics of the auditory stimulus, and quantification of behavioral responses), a striking resemblance in development of localization between early (this study) and late bilateral implantation existed (Figure 4, panel B). The rate of development was identical, whereas intercepts differed slightly (late cochlear implants: slope = 0.21/year, intercept = 0.79; early cochlear implants: slope = 0.21/year, intercept = 0.82).

### 3.4. Reliability of Sound Localization Measurements

The 95% C.I. of the test–retest differences (−0.098 to 0.037) included 0, reflecting that no significant learning effect occurred. The 95% C.I. for the EI for a single measurement was ±0.11 (*n* = 42). The test–retest differences did not depend on the number of obtained responses during a test (r = 0.13, *p* = 0.40).

## 4. Discussion

We found that infants with bilateral severe-to-profound congenital hearing loss develop horizontal sound localization abilities after bilateral cochlear implantation. When contrasting data from the current study with previous data from infants with normal hearing, we found that the developmental rates in these groups were similar. While it seems unlikely that localization will reach the same accuracy as in normal hearing based on CI studies in adults [31], the rate of improvement emphasizes the need of early provision of hearing in both ears for children with severe-to-profound hearing loss to allow development of spatial hearing near ages for which development normally occurs. In addition to being a safety matter in for example traffic, adults with hearing loss report that difficulties in localization of sounds are associated with loss of concentration, confusion of sounds, and a wish to escape settings in which this occurs [32]. Additionally, accurate localization is likely to improve communication since audiovisual cues are important for speech understanding when hearing loss is present [33]. Less is known about how impaired sound localization during infancy affects learning and interaction in daily life and should be targeted in future research. For children with unilateral hearing loss, a condition that typically is associated with impaired localization [34], it is 10 times more common having to repeat at least one year in school [35].

When contrasting the current data with previous data from children with relatively late cochlear implantation, we found a striking resemblance between infants and children in early school-age in the rate of development following implantation. This demonstrates that a sensitive period for human spatial hearing is not restricted to early development, corroborating findings in adult humans [6] and ferrets [16] who adapt their behavior to altered spatial cues. Results differ from barn owls [36] and mice [37], for which age limits sensitive periods for development of sound localization or the binaural cues it is based on. In children implanted bilaterally after 5 years of age, localization performance is poor one year after implantation [38], but data on long-term localization performance for late bilateral implantation is unknown. It is noteworthy that the high similarity in the rate of development and between-subject variability of sound localization abilities found in the present study between younger and older children with cochlear implants occurred despite methodological differences in how localization accuracy was measured. Infants’ responses in the present study were obtained through eye-gaze, whereas older children with implants [23] pointed at or verbally indicated the perceived sound-source azimuth. In addition, children in the present study listened to a continuous sound, whereas older children listened to sounds of relatively short duration. For both groups, between-subject variability in localization accuracy was high and time since activation of BiCI explained the same amount of variance in localization accuracy (25%). The underlying causes for variability in binaural hearing in individuals with implants have been targeted in recent years [39,40,41,42,43], revealing etiology of the hearing loss, duration of hearing loss, and surgical procedures and subsequent bilateral fitting of sound processors as variables that may affect results. Importantly, while current and previous data presented together here show that the developmental rate of localization accuracy is comparable for children with normal hearing and cochlear implants, localization performance after prolonged cochlear implant use does not reach that of individuals with normal hearing [17,44,45]. One reason for less accurate localization despite many years of cochlear implant stimulation is that thresholds for important cues underlying accurate localization (interaural level and time differences) typically are substantially worse for listeners with cochlear implants compared to normal hearing [46], owing to technical and surgical limitations (see, e.g., ref. [47] for a discussion). Future studies including long-term follow up of children who received bilateral cochlear implants as infants may reveal if localization performance plateaus at higher accuracy than later implanted children, and if early localization abilities have implications for, e.g., learning, language and social interaction. Factors of interest in such future studies should be to determine underlying causes for between-subject variability through genetic testing (≈50% of congenital sensorineural hearing loss are genetic in origin [48,49]) and radiological investigation of bilateral cochlear implant electrode placement to assess the impact of interaural frequency mismatch which negatively affects binaural hearing [39].

A limitation of the comparison between data collected in the current study and data from children with NH from previous work is that previous data are cross-sectional and based on a relatively small sample. However, the data from children with NH should be representative given previously performed analyses [28] of localization accuracy in larger samples of children with NH aged 0.5 to 1.5 years (*n* = 80) showing a rate of improvement similar to what was found in our smaller sample [28,50].

Data presented here suggest an intrinsic mechanism for the development of horizontal sound localization abilities. Our study improves on previous work on spatial hearing in children with cochlear implants [17,23,51,52] due to its inclusion of children at a very young age and its longitudinal follow-up before school-age. As long as cochlear implantation may be performed safely in infants, our findings suggest that implantation should occur as early as possible to allow development of spatial hearing near ages for which development normally occurs.

## Figures and Tables

**Figure 1 jcm-11-06758-f001:**
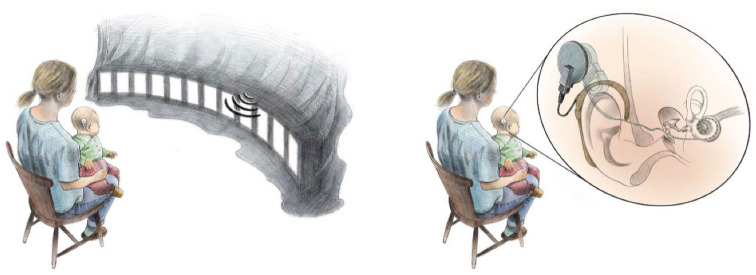
Experimental setup for determination of horizontal sound localization accuracy in infants and young children listening through bilateral cochlear implants. The left panel illustrates the position of the child relative to an array of loudspeaker/display-pairs. Loudspeakers were covered in black cloth to attract the child’s gaze to the visual displays. A continuous auditory-visual stimulus was presented from a loudspeaker/display-pair and randomly shifted in azimuth. Simultaneously with an azimuthal shift, the visual part of the stimulus was stopped for 1.6 s and eye-gaze patterns in response to the auditory stimulus were recorded before the visual part of the stimulus returned. The right panel illustrates the implanted and external parts of a cochlear implant system. An array of electrode contacts resides in the cochlea, stimulating the auditory nerve. The electronics of the cochlear implant are driven by an external sound processor behind the ear. Illustration by Mats Ceder.

**Figure 2 jcm-11-06758-f002:**
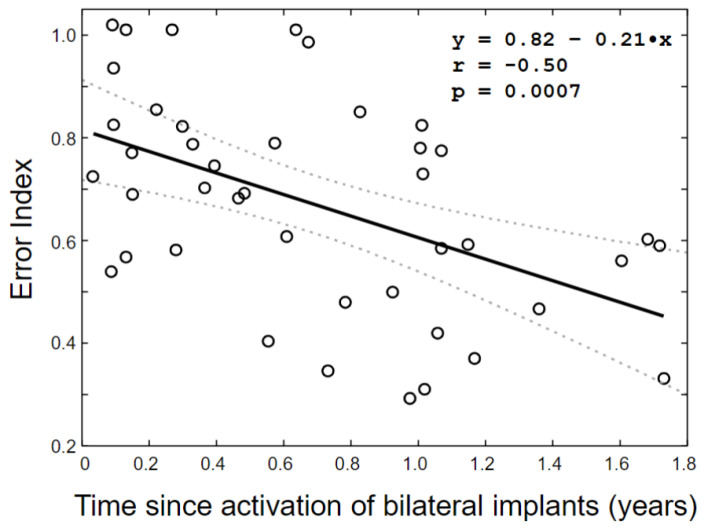
Children with congenital severe-to-profound hearing loss develop horizontal sound localization abilities with increasing time since activation of bilateral cochlear implants. The black open circles depict localization accuracy (Error Index) for 18 infants measured at 1 to 5 occasions (*n* = 42, mean follow-up time = 0.9 years (0.09 years–1.7 years); mean age at test = 1.5 years (0.58 years–3.6 years). Localization accuracy increased as a function of time since activation of bilateral cochlear implants (R^2^ = 0.25, F = 13.6, *n* = 42, *p* = 0.0007, linear mixed model). The black solid line is the linear fit from a linear mixed model analysis, and the dotted lines depict the 95% confidence interval of the fit.

**Figure 3 jcm-11-06758-f003:**
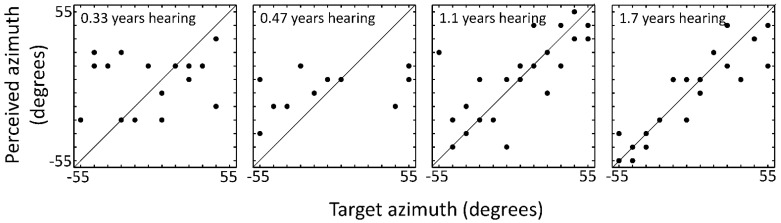
Perceived as a function of target sound-source azimuth at 4 occurrences (one per panel) for an individual child. In the left panel, this child had bilateral cochlear implants activated for 0.33 years. As experience with bilateral cochlear implants increases (from left to right), datapoints approach the line of equality corresponding to perfect sound localization accuracy in this task.

**Figure 4 jcm-11-06758-f004:**
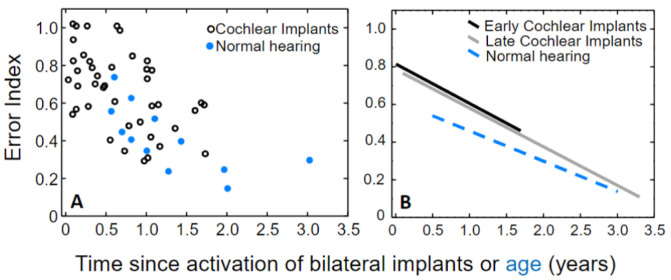
(**A**): The black open circles depict localization accuracy in children with bilateral cochlear implants, and the blue filled circles depict localization accuracy in children with normal hearing from Asp et al. (2016) [28]. (**B**): The lines show linear fits based on data from the present study in infants (black) and previous data from children with normal hearing (blue, Asp et al. (2016) [28]) and children with relatively late sequential bilateral implantation (grey, Asp et al. (2011) [23]).

**Table 1 jcm-11-06758-t001:** Background data on the children who participated in the study. Children are sorted in ascending age order. Two of twenty included children did not cooperate to sound localization testing and are not shown.

Age at Implantation (Years)	Cochlear Implant Model	Sound Processor	Sex	Etiology
0.42	CI522	CP1000	M	Connexin 26
0.45	CI522	CP1000	M	Genetic testing performed; no mutation found
0.45	Synchrony 2 Flex 28	Sonnet 2	M	Cause not investigated
0.47	CI612	CP1000	F	Cause not investigated
0.48	CI532	CP1000	F	Genetic testing performed; no mutation found
0.51	Synchrony 2 Flex 28	Sonnet 2	F	Unknown; no positive cCMV infection found
0.51	CI612	CP1000	M	Connexin 26
0.55	CI522	CP1000	F	Connexin 26
0.56	Synchrony 2 Flex 28	Sonnet 2	F	Connexin 26
0.58	CI522	CP1000	F	cCMV
0.58	CI512	CP1000	M	cCMV
0.65	CI612	CP1000	M	Cause not investigated
0.76	Synchrony 2 Flex 28	Sonnet 2	F	Genetic testing performed; uncertain causative gene mutation
0.91	CI612	CP1000	M	Genetic testing performed; no mutation found
1.0	Synchrony 2 Flex 28	Sonnet 2	M	Connexin 26
1.6	CI612	CP1000	M	Cause not investigated
1.8	CI522	CP1000	M	Unknown; no positive cCMV infection found
2.3	CI612	CP1000	F	Cause not investigated

CI522, CI512, CI532, and CI612; cochlear implants manufactured by Cochlear Corporation, Sydney, Australia. Synchrony 2 Flex28; cochlear implant manufactured by Med-El GmbH, Innsbruck, Austria. CP1000; sound processor manufactured by Cochlear Corporation, Sydney, Australia. Sonnet 2; sound processor manufactured by Med-El GmbH, Innsbruck, Austria. M; Male. F; Female. cCMV; congenital cytomegalovirus.

## Data Availability

The data presented in this study are available on request from the corresponding author. The data are not publicly available due to ethical, legal and privacy issues.

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
