# Peer review of "Development of Sound Localization in Infants and Young Children with Cochlear Implants"

_jcm, 2022, doi:10.3390/jcm11226758_

Round 1
Reviewer 1 Report
In their manuscript "development of sound localizaton in infants and young children with cochlear implants" by Asp et al. longitudinal data on 20 children with mainly early bilateral cochlea implantation are presented focusing on the rate of improvement of horizontal localization accuracy. Data are compared to a historical control group including children with normal hearing and children with late implantation.
The introduction is comprehensive, the studies aims are clear, methods are described precisly and the results are described adequat. The discussion is again comprehensive, the main results are discussed appropriate and the conclusions are supported by the data.
Overall this is an intersting and important study. Despite its low n and some minor points.
Minor points:
1) Introduction page 2, lines 46-50. The importance of family centered early intervention for language development should be mentioned.
2) Study subjects: any data on etiology?
3) Study subjects: any data on vestibular involvement? Would be important in view of the often decresed visual localization speed in these children.
Author Response
Reply: thank you for your efforts in reviewing this paper.
Minor points:
1) Introduction page 2, lines 46-50. The importance of family centered early intervention for language development should be mentioned.
Reply: Mentioned as suggested (line 51)
2) Study subjects: any data on etiology?
Reply: Table 1, added after suggestion by another reviewer, shows available data on etiology.
3) Study subjects: any data on vestibular involvement? Would be important in view of the often decresed visual localization speed in these children.
Reply: While we have data on vestibular function in many cases, we did not seek ethical approval to report these data.
Reviewer 2 Report
The authors wrote an article about the development of sound localization in infants and young children with cochlear implants. The article is interesting, well written, the topic is not so new, the concept was already described in literature. There are some suggestions to increase the scientific quality of the manuscript and to give a better impact in scientific literature.
1. In the study design, please talk about the kind of surgery and the possibility to use soft surgery to give a better audiological improvement. Please use this reference: Freni F, Gazia F, Slavutsky V, Scherdel EP, Nicenboim L, Posada R, Portelli D, Galletti B, Galletti F. Cochlear Implant Surgery: Endomeatal Approach versus Posterior Tympanotomy. Int J Environ Res Public Health. 2020 Jun 12;17(12):4187.
2. Please insert a table with the study population (gender, age, kind of cochlear implant...)
3. In the discussion, in the section of limits or future study, insert the possibility to compare sound localization in children with bilateral CI, but the second one implanted late that 4 years.
Author Response
Thank you for your efforts in revising this manuscript.
- In the study design, please talk about the kind of surgery and the possibility to use soft surgery to give a better audiological improvement. Please use this reference: Freni F, Gazia F, Slavutsky V, Scherdel EP, Nicenboim L, Posada R, Portelli D, Galletti B, Galletti F. Cochlear Implant Surgery: Endomeatal Approach versus Posterior Tympanotomy. Int J Environ Res Public Health. 2020 Jun 12;17(12):4187.
Reply: The effect of surgical procedures on binaural hearing with cochlear implants are relevant and is mentioned in lines 272-276.
- Please insert a table with the study population (gender, age, kind of cochlear implant...)
Reply: A table (Table 1) is included as suggested.
- In the discussion, in the section of limits or future study, insert the possibility to compare sound localization in children with bilateral CI, but the second one implanted late that 4 years
Reply: We added a discussion and two references on this topic in the revised manuscript (lines 261-263).